# Electrochemical and Tribocorrosion Study of D2 Steel Coated with TiN with C or Cr Addition Films in 3.5 wt% of NaCl in Bi-Distillated Water Solution

**DOI:** 10.3390/ma18122733

**Published:** 2025-06-11

**Authors:** Ernesto David García-Bustos, Diego Maxemin-Lugo, Norberto Diez-Torres, Noé López-Perrusquia, Marco Antonio Doñu-Ruiz, Martin Flores-Martinez, Johans Restrepo, Stephen Muhl-Saunders

**Affiliations:** 1Secretaria de Ciencia, Humanidades, Tecnología e Inovación, Universidad Politécnica del Valle de México, Tultitlan 54910, Edo. México, Mexico; 2Instituto Tecnológico y de Estudios Superiores de Occidente, Tlaquepaque 45604, Jalisco, Mexico; d_maxemin@hotmail.com (D.M.-L.); forzadiez@hotmail.com (N.D.-T.); 3Grupo Ciencia e Ingeniería de Materiales, Universidad Politécnica del Valle de México, Tultitlan 54910, Edo. México, Mexico; marco.donu@upvm.edu.mx; 4Centro Universitario de Ciencias Exactas e Ingenierías, Universidad de Guadalajara, Guadalajara 44430, Jalisco, Mexico; martin.fmartinez@academicos.udg.mx; 5SADOSA S.A. de C.V., Aragón, Ciudad de México 07000, Mexico; johansrestrepo@gmail.com; 6Instituto de Investigaciones en Materiales, Universidad Nacional Autónoma de México, Ciudad de México 04510, Mexico; muhl@unam.mx

**Keywords:** D2 steel, cathodic arc, titanium nitride, corrosion, tribocorrosion

## Abstract

Food security is one of the main problems in several countries. In food processing the cutting operation is very important as the operation is basic to food preparation. Due to cutting tools being exposed to a high-demand environment that includes high contact pressure, a corrosive atmosphere, and a high-speed process, they are subject to high mechanical and corrosive wear that reduces their lifetime and efficiency. Tribocorrosion is one of the main phenomena that reduces the lifetime and efficiency of cutting tools. This work presents electrochemical and tribocorrosion studies of D2 steel surfaces coated with TiN, TiCN, and TiCrN films. The samples were coated by a commercial source, using the PVD-cathodic arc technique. The crystalline structure of TiN and TiCN films presented a TiN and Ti phase, while the crystalline structure of TiCrN showed CrN and Cr phases. The films exhibited good adhesion, but the surfaces coated with TiN and TiCN films presented lower hardness. Although the TiN, TiCN, and TiCrN films showed better wear and corrosion resistance than the D2 steel surfaces, the inclusion of C and Cr in the TiN films decreased the TiN wear and electrochemical resistance in 3.5% (*w*/*w*) of NaCl solution.

## 1. Introduction

Cutting tools used in home kitchens are typically made of stainless steel, as this material offers excellent corrosion resistance along with sufficient mechanical and tribological properties for low-demand work operations. However, the food industry processes materials with varying levels of complexity, including low, medium, and high complexity, each possessing different elastoplastic and viscoelastic properties [1,2]. An ongoing challenge in the food industry is how to enhance the efficiency of food processing to help mitigate the food crisis. The Food and Agriculture Organization (FAO) reported in 2023 that approximately 58 million people in 53 countries were food insecure [3]. A crucial factor in food processing is the cutting velocity, which varies based on the complexity of the materials being processed. This variation impacts edge deformation, fracture, and the friction force generated during sliding contact between the cutting tool and the food material [4,5,6]. Corrosion is one of the most important issues that reduces the life-time of the tools and infrastructure, with stainless steel and aluminum alloy mostly used as the metallic components in the food industry [7,8]. However, there are operations that require high wear and stress resistance, such as the cutting operation. Examples of such demands occur in the fisheries and aquaculture industry, that is among the most significant sectors in the food industry [5], vegetable and fruits processing [9,10], the meat industry [11] and other food sectors.

Shear blades are commonly used to cut various materials in the food industry, making the sliding process one of the most prevalent methods in food manufacturing. Typically, the blades are made from tool steel. Bremer et al. [12] noted that an effective slicer blade must strike a balance between cost, manufacturability, weight, sharpness, and rigidity to achieve efficiency in high-speed cutting applications [13]. D-class steel is favored across several industries due to its excellent mechanical and wear-resistant properties. D2 steel, which contains a high percentage of carbon and chromium (approximately 12% chromium), is commonly used for mold making, and forming dies, cold drawing punches, shear blades, and cutting tools [14,15,16,17]. While the chromium content provides D2 steel with mild corrosion resistance, slicer blades often operate in demanding environments characterized by a combination of high-speed operation, corrosive conditions, and high fatigue levels. These factors can significantly damage cutting tools [5,12,13,17]. The interplay between mechanical wear and corrosion can significantly enhance the wear of surfaces in contact during relative motion, a phenomenon known as tribocorrosion [18,19,20,21,22,23]. Enhancing surface properties has proven to be a crucial strategy in improving the efficiency of components subjected to tribological operations in various environments, including those impacted by tribocorrosion. In this regard, modification of AISI D2 steel surfaces in order to improve corrosion, wear, and tribocorrosion properties has been studied in several projects. For example, Kaigude et al. [24] reported the use of Electrical Discharge Machining (EDM) used like a matching process; Reséndiz-Calderón et al. [25] reported increment in the wear properties of D2 steel due to the application of a boriding thermal process; Castillejo et al. [26] reported increment in the wear and corrosion property of D2 steel using the thermos-reactive process to deposit a chromium–vanadium carbide film; Voglar et al. [27] used cryogenic treatment to modify the hardness and corrosion properties of D2 steel surfaces.

One of the most effective methods for modifying surface properties is the deposition of films with specific characteristics. The PVD-cathodic arc process is an efficient industrial technique for applying films, offering precise control over their characteristics [28,29,30]. TiN (titanium nitride) film has found extensive industrial application due to its ability to improve mechanical, corrosion, and tribological properties, making it a suitable protective layer for systems exposed to tribocorrosion [31,32,33]. Furthermore, ternary materials have been integrated into the TiN matrix to enhance the specific properties further.

Addition of carbon (C) to the titanium nitride (TiN) matrix has been employed to enhance the mechanical and tribological properties of TiN films. This process results in the formation of a TiN matrix that incorporates Ti-C, C-C, or C-N phases, which leads to improved hardness and wear resistance, as well as reduction in the coefficient of friction (CoF) in certain tribological systems [34,35,36,37,38]. Chromium (Cr) is commonly used in various alloys and films to enhance their corrosion resistance by forming a stable chromium oxide (Cr_2_O_3_) layer [39,40,41]. This protective layer is used in D2 tool steel, where the chromium oxide offers moderate protection to the steel surfaces. However, this oxide layer is inadequate for safeguarding D2 steel surfaces against tribocorrosion phenomena [14,15,17].

This study presents the electrochemical and tribocorrosion results for D2 steel surfaces coated with TiN, TiCN, and TiCrN films produced by the PVD-cathodic arc technique, using a commercial industry process in a 3.5 wt% NaCl solution in bi-distilled water. The TiN and TiCN films significantly improve the electrochemical and tribocorrosion properties of the D2 steel substrate, resulting in lower values of the corrosion potential (Ecorr) and corrosion current density (Icorr), high percentage protection (Pi%), and low wear rate. However, the TiCrN film exhibited lower electrochemical performance than uncoated D2 steel, showing a higher corrosion current (Icorr) and higher coefficient of friction (CoF), but with higher wear resistance than D2 steel surfaces.

This work shows the quantitative relationship between deposition process parameters and coating performance. The development of a coating by the PVD-cathodic arc technology influences the mechanical, electrochemical, and tribological properties. The quantitative results obtained of the correlations between the deposition parameters and the resulting properties of TiN, TiCN, and TiCrN coatings on D2 steel are presented in the manuscript data. This can inform the design of coatings that are suitable for particular industrial environments, as described by Xian et al. [42] and Du et al. [43].

## 2. Materials and Methods

### 2.1. Surface Preparation

D2 steel substrates were obtained of 2-inch (50.8 mm) diameter with a thickness of 5 mm. The substrates were polished using various grit sandpapers and diamond paste to achieve a high-quality surface finish. Before deposition, the substrates were cleaned using an industrial soap and water solution, followed by rinsing with acetone and ethyl alcohol. The TiN film containing carbon and chromium was deposited using the PVD-cathodic arc technique from a commercial source. Ti and Cr targets, each with a purity of 99.99%, were utilized along with N_2_ and C_2_H_2_ gases as precursors for producing nitride, carbide, and carbonitride films. To produce the TiN films, four titanium targets were employed, operating at 85 A in a N_2_ atmosphere with a working pressure of 3.5 Pa and a bias voltage of 250 V. For the TiCN film, the same parameters used for the TiN film were applied, with the addition of 10% C_2_H_2_ in the N_2_ gas mixture. For the TiCrN film, the same parameters as those for the TiN film were used, but with three chromium targets, also operating at 85 A. The deposition process was carried out at room temperature (around 273 K).

### 2.2. Surface Characterization

The coated and uncoated D2 surfaces were characterized using several techniques, including X-ray diffraction (XRD) with an Empyrean Panalytical diffractometer (Enigma Business Park, Grovewood Road, Malvern. WR14 1XZ, United Kingdom) utilizing a copper source in a Bragg–Brentano configuration, energy-dispersive X-ray spectroscopy (EDS) with a Jeol system, scanning electron microscopy (SEM) (JEOL Ltd., 3-1-2 Musashino, Akishima, Tokyo) with a Jeol microscope, and Raman spectroscopy using a 532 nm laser source with a thermo-confocal Raman spectrometer (168 Third Avenue, Waltham, MA, USA). These techniques were employed to determine the crystalline structure, elemental composition, and morphology of the surfaces.

### 2.3. Surface Properties

#### 2.3.1. Mechanical Properties

The hardness of D2 surfaces, both coated and uncoated, was measured using a Future-Tech micro-indenter at applied loads of 300, 100, 50, and 25 g, with hardness values assessed using the optical microscopy of the equipment. The fracture toughness of the layers was determined using the Vickers indentation test at an applied load of 2 kg. The scratch tests were carried out using a spherical pin of Al_2_O_3_ of 1/16 inch of diameter from 0 to 40 N, with a track length of 5 mm at 5 mm/min speed (ASTM C1624 [44]). The indentation marks and scratch tracks were analyzed through SEM microscopy.

#### 2.3.2. Electrochemical Properties

Electrochemical testing was conducted using a customized electrochemical cell featuring a saturated calomel electrode (SCE) as the reference electrode, a cylindrical carbon electrode, the D2 samples as the working electrode, and a potentiostat (Reference 600, Gamry). The potentiodynamic tests were performed in 30 mL of a 3.5 wt% NaCl solution in bi-distilled water at a scanning speed of 2 mV/s within a voltage range of −1 to 1 V, and the current (in A/cm^2^) was measured during the potentiodynamic test. Prior to the potentiodynamic tests, a 1 h passivation period was observed (this standard time was determined from the stabilization time of the OCP voltages for the samples). During this time, the Open Circuit Potential (OCP) was measured. All the tests were carried out at the environmental temperature (around 278 K).

Tribocorrosion tests were conducted using a UTM2 CETR tribometer with a reciprocating sliding motion system. The tests involved a go-back period of one second with a 10 mm race track for 0.5 h (36 m of sliding distance) while applying a load of 1 N. A ZrO_2_ ball with a 4.9 mm diameter served as the counter body in these tests. The samples were immersed in 30 mL of a 3.5 wt% NaCl solution in bi-distilled water, utilizing a saturated calomel electrode (SCE) as the reference electrode, a cylindrical carbon electrode, and the D2 samples as working electrodes. Measurements of the Open Circuit Potential (OCP) and friction force (Ff) were recorded during the tribocorrosion tests. These test parameters were selected to ensure that the cutting operation was consistent with that used in the food industry with a Hertzian contact pressure of 620 Mpa (ZrO ball of 3.9 mm on contact with TiN surfaces) [45,46,47]. All the tests were carried out at room temperature. The wear tracks were analyzed using optical microscopy and an optical profilometer. The wear rate was calculated using the following formula:$$\mathrm{wear}\;\mathrm{rate}=\frac{V\;\left(m^{3}\right)}{L\;\left(m\right)F\left(N\right)}$$
where V is the wear volume (ASTM 133 [48]), L is the sliding distance, and F is the applied force [49].

## 3. Results

### 3.1. Surfaces Characteristics

The thicknesses of these films were 1.9 ± 0.33 μm, 2.5 ± 0.22 μm, and 2.15 ± 0.33 μm for the TiN, TiCN, and TiCrN films, respectively. The surface roughness (Rq) value of D2 steel increased as the films were deposited, rising from 0.03 ± 0.01 μm for the substrate value to 0.17 ± 0.02 μm, 0.31 ± 0.03 μm, and 0.49 ± 0.03 μm for the TiN, TiCN, and TiCrN films, respectively. This increase in surface roughness is commonly observed when using the PVD-cathodic arc deposition technique, as macroparticles (Mp) expelled from the target by the high current arc during the deposition process settle on the substrate alongside the deposited atoms. In the same way, porosity (P) was observed on the surface that could be caused by macroparticles spallation (see Figure 1) [28,29,30]. Figure 1 shows that the inclusion of C and Cr in the TiN film increased the roughness due to increment of macroparticles and porosity of the layers, with higher defects for the TiN and TiCrN films. The increase in porosity and macroparticles in TiN, due to the addition of carbon (C) and chromium (Cr), can be attributed to a chemical reaction between the metallic target (titanium, Ti) and the reactive atmosphere containing nitrogen (N) and carbon. This reaction leads to the formation of titanium carbide (TiC) and titanium nitride (TiN) on the surface of the targets in the deposition of TiCN films or TiN and CrN in the deposition of TiCrN films, increasing the melting point of the surfaces and spallation of macroparticles during the evaporation process, which then deposit onto the substrate [50,51,52].

The elemental composition of the TiN film consists of 70 atm% titanium (Ti) and 30 atm% nitrogen (N). The crystalline structure of the TiN layer combines the FCC-TiN phase, which includes the (111), (200), and (222) planes (PDF# 65-0565), with the α-Ti phase characterized by the (100), (002), (101), (103), and (201) planes (PDF# 65-6231), as well as the α-Fe phase from the steel substrate, represented by the (110) and (200) planes [14,16,35,53]. This combination of phases corresponds to the elemental composition, highlighting the presence of free Ti metallic in the TiN film. The Raman spectra of the TiN film exhibits several vibration modes: The transversal (TA) mode at 204 cm^−1^, longitudinal (LA) mode at 289 cm^−1^, and a second mode of the acoustical (2A) vibration at 434 cm^−1^, in addition to the transversal optical (TO) mode at 549 cm^−1^, longitudinal optical (LO) mode at 621 cm^−1^, and A+O mode at 817 cm^−1^. These bands have been extensively reported in studies focusing on the film characteristics, such as the crystalline structure and elemental composition (see Figure 2) [35,54,55,56,57,58]. The Raman spectra of TiN also show a band at 1106 cm^−1^ that is not usually reported but Spengler et al. [58] reported that this band corresponds to the 2O vibrational mode. The elemental composition of the TiCN film comprises 38 atm% Ti, 45 atm% N, and 17 atm% carbon (C). Its crystalline structure resembles that of the TiN film, showing the (111), (200), and (220) planes of the cubic TiC phase (see Figure 2a) [59,60]. The Raman spectra of the TiCN film display characteristics of the TA, LA, and 2A modes from the TiN Raman spectra, along with the D band at 1345 cm^−1^ and the G band at 1574 cm^−1^ (see Figure 2b) [61,62]. The elemental composition and the presence of both NaCl-FCC structures of the TiN and TiC phases suggest the existence of C-N bonds (predominantly sp^3^ bonding in ta-C:N [62]) and C-C bonds within a graphitic amorphous structure. The TiCrN film is composed of 34 atm% Ti, 30 atm% N, and 36 atm% chromium (Cr). The TiCrN film possesses a multi-crystalline structure that includes the TiN phase along with the (111), (200), and (220) planes of the CrN (FCC) phase and the (110) and (200) planes of the Cr (BCC) phase and the same band of TiN film in its Raman spectra (see Figure 2) [40,63,64].

### 3.2. Mechanical Properties

#### 3.2.1. Hardness Test

Figure 3a illustrates the hardness (HV) of D2 steel coated with TiN, TiCN, and TiCrN at various applied loads of 300 g, 100 g, 50 g, and 25 g. The hardness values were notably recorded with an applied load of 25 g, as the indentation marks produced at this load had less impact on the substrate. The highest hardness value obtained for D2 steel was 7.5 ± 0.3 GPa. The hardness values measured for the coatings TiN, TiCN, and TiCrN were 16.1 ± 2.7 GPa, 14.2 ± 1.2 GPa, and 18.6 ± 3.7 GPa, respectively. While these values are comparable to some previously reported hardness values for TiN films, they fall within the lower range of those reported for TiN [65,66,67,68,69]. This variation can be attributed to the fact that during the application of load, the contact stresses exceed the thickness of the films. This leads to a combination of plastic deformation of the coating and material removal. Evidence of this plastic deformation was observed through the contact depth values obtained during the microhardness Vickers indentation test at a load of 25 g, where the contact depths were found to be 0.76 ± 0.06 μm, 0.8 ± 0.02 μm, and 0.71 ± 0.08 μm for the TiN, TiCN, and TiCrN films, respectively, exceeding more than 10% of the films’ thickness [68,69,70,71,72]. In order to obtain a theoretical approximation value of the films, the Korsunsky mathematical model was used to determinate the hardness of the films [73,74,75]:$$H_{c}=H_{s}+\frac{H_{f}-H_{s}}{1+{k\beta }^{2}}$$
where H_c_ is the composite hardness, H_f_ is the film hardness, H_s_ is the substrate hardness, k is a fitting parameter (in this case, the k value reported by Tuck et al. [73] for the TiN film deposited by the catholic arc technique was used), and β is defined by Korsunsky et al. [75] as the indentation depth normalized with respect to the coating thickness.

The hardness values obtained with the model were 31.9 ± 5.2 GPa, 23.2± 2.7, and 32.2 ± 5.7 GPa for TiN, TiCN, and TiCrN films, respectively. These hardness values are more similar to the reported hardness value for TiN film [65,73,76,77]. Although the TiCrN film contains a Cr metallic phase, its hardness value is higher than that of the TiN and TiCN films due to the reduction in the Ti metallic phase presence [78,79,80,81,82]. The films demonstrated acceptable fracture toughness, as indicated by the indentation marks produced during the tests at a 2 kg load. There were no fractures or detachments observed at the corners of the films, suggesting that the films possess a high level of plastic deformation that effectively dissipates stress. This plastic deformation is attributed to the metallic phases present in the films (see Figure 3b) [83,84,85].

#### 3.2.2. Scratch Test

Figure 4 presents the wear track zones of the scratch of D2 steel coated with TiN, TiCN, and TiCrN films. The scratch track on the TiN film presented plastic deformation (Pd) and fractures (Fl) on the border of the wear track. Nevertheless, there was no spallation of the film, showing good adhesion to the substrate. The fractures of the layer were presented at 21 N of applied load, this being an L1 critical load of the scratch test. The TiCN layer showed plastic deformation halfway through the track, but at the end of the track, the layer presented fractures (Fl) and spallation (Sl), showing the L1 and L2 caption was updated critical load at 36 N and 38 N, respectively. Between the TiN and TiCN films, the TiCrN layer showed fractures and spallation at 25 N (L1 and L2) with plastic deformation on the border of the track. The direction and form of the fractures of the TiCrN film failed by tension stress [86,87]. Although the TiCN and TiCrN films presented spallation during the scratch tests, these films exhibited plastic behavior because the TiN, TiCN, and TiCrN films showed high plastic deformation during the scratch tests.

### 3.3. Electrochemical Properties

Figure 5 illustrates the open circuit potential (OCP) and potentiodynamic polarization curves (PPC-Tafel curves) from the corrosion tests on D2 steel surfaces coated with TiN, TiCN, and TiCrN films in a 3.5% NaCl solution. The D2 steel surfaces exhibited lower OCP values with some deformation due to the corrosion process after 1500 s. The coated D2 steel surfaces with TiCN and TiCrN showed lower OCP values compared to those coated with TiN, indicating that the incorporation of carbon (C) and chromium (Cr) into the TiN matrix diminished its corrosion resistance. The PPC graph for the D2 steel surfaces displayed active, passivation, and trans-passivation regions, with values of Ecorr = −791 mV and Icorr = 3.66 × 10^−1^ μA/cm^2^ (see Table 1). The surfaces coated with TiN and TiCN films demonstrated a more impressive performance than the uncoated D2 steel, with values of Ecorr_TiN_ = −212 mV and Icorr_TiN_ = 1.17 × 10^−4^ μA/cm^2^, and Ecorr_TiCN_ = −328 mV and Icorr_TiCN_ = 1.16 × 10^−2^ μA/cm^2^ (see Table 1). These results indicate that the TiN film significantly reduced the corrosion effects on the D2 steel surfaces, achieving a protection percentage (Pi%) of 99.97% [18,22,23,88]. However, the protection was reduced to 96.82% with the inclusion of carbon due to the higher presence of the Ti metallic phase in the film that reacted with the corrosive environment during the pitting process [89,90,91]. Despite the Ecorr_TiCrN_ being higher than that of the D2 steel substrate, the inclusion of chromium in the TiN matrix notably decreased the protection percentage, resulting in a higher corrosion current than the substrate, with Icorr_TiCrN_ = 5.17 × 10^−1^ μA/cm^2^. Figure 5c shows examples of the wear marks caused during the PPC tests on the D2 steel substrate and the TiCN- and TiCrN-coated surfaces. The D2 steel surfaces exhibited pitting, which resulted in porosity and a granular morphology formed by the corrosion products. The TiCN film displayed more significant corrosion marks than the D2 steel surfaces. The pitting on the TiCN film indicated wear of the film, leading to a columnar morphology. The corrosion marks on the TiCrN films revealed nucleation points that caused lamellar detachment of the film, exposing the substrate beneath. In Figure 1 and Figure 3, it is shown that the macroparticles and porosity increased with the addition of C and Cr in the TiN films, reducing the electrochemical properties of the TiN layer. This reduction was produced due to the deposition process of the films with the PVD-cathodic arc technique; hot macroparticles were deposited and during the cooling process, this reduced their size, causing cracks on the macroparticle borders with the layers that permit diffusion of the corrosive solution. As a result, the TiCN and TiCrN films presented a lower corrosion resistance than the TiN film. Although some works have reported increase in the corrosion resistance of the binary layer (metallic-carbide MC or metallic-nitride MN films) with the addition of a ternary material such as Cr and C, defects and the morphology of the films can reduce this performance [92,93,94]. The addition of carbon (C) and chromium (Cr) has been utilized to enhance the mechanical and electrochemical properties of materials, particularly with respect to Cr. The inclusion of chromium can lead to the formation of a Cr_2_O_3_ film, which exhibits high chemical and mechanical stability [48,95,96,97]. However, these findings indicate a decrease in the corrosion resistance of the TiN film when C and Cr are added. This reduction can be attributed to the increased filtration of corrosive solutions through the porous material and defects present in the TiN matrix, which are caused by macroparticles and variations in the composition generated during the deposition process [89,96,97,98]. In the same way, the porosity and defects in the TiCN and TiCrN films increased the exposed area to the corrosive solution, increasing the current density (Icorr). This effect can be observed in the variation in the Pi value, where the TiCN and TiCrN films presented a lower Pi value than the TiN film, especially for the Pi value of the TiCrN film, where increment in the area improved the redox process, causing a higher ion liberation that produced a higher Icorr value on the coated D2 steel surfaces with TiCrN film than the uncoated D2 surfaces. These effects can be observed in the SEM images of the corrosion on the D2 steel surfaces uncoated and coated with TiCN and TiCrN films (see Figure 5c). It was observed that the corrosion wear marks on the D2 steel presented a granular morphology that was produced by the corrosion process on the grain borders. The corrosion marks observed in the TiCN film displayed a columnar morphology, which is likely produced by the filtration of material within the TiN matrix along the edges of the film’s columnar structure. Similarly, the corrosion marks in the TiCrN film indicate a comparable filtration process occurring through its columnar morphology. However, the corrosion marks on the TiCrN film also exhibit delamination and spallation in certain layers, suggesting that the filtration occurred in both columnar and transversal directions.

### 3.4. Tribocorrosion Properties

#### 3.4.1. Wear

Figure 6 presents the wear track and wear profiles resulting from tribocorrosion tests conducted on D2 steel coated with TiN, TiCN, and TiCrN layers. The wear track on the D2 steel substrate displayed a soft surface, lacking plastic deformation or transferred material, which is common in the wear tracks of metallic surfaces under dry conditions. Small abrasive marks were noted on the wear track, with debris accumulating at its borders. On the D2 steel surfaces coated with a TiN layer, the wear track exhibited abrasive marks along its length, with debris present at the center and along the borders of the worn zones. The wear track resulting from tribocorrosion tests on the TiCN layer showed smoother worn surfaces than the TiN layer, with debris located at the borders and some areas covered by a tribolayer formed from the accumulated debris. The wear track on the TiCrN film also demonstrated abrasive marks and debris on its worn surfaces. In this case, the debris accumulation formed a tribolayer with a soft morphology in the center of the wear track, along with plastic deformation and fractures observed at the track’s borders. The wear profiles revealed that the D2 steel surfaces exhibited the highest wear rate, followed by the TiCrN layer, which, while showing a lower wear track depth than the TiN and TiCN films, had a greater width. Similarly, the wear track on the TiCN film had a shallower depth than that of the TiN film but with a greater width. These findings suggest that although the wear tracks on the TiCN and TiCrN layers had lower depths than the TiN film, they provided better protection against tribocorrosion. This performance is due to the smaller width of the wear tracks produced during the tribocorrosion tests on the TiN film compared to those on the TiCN and TiCrN films. The wear tracks observed on the counter bodies displayed adhered material along the borders due to plastic deformation resulting from the fatigue process. In particular, the wear track of the ball used in the tribocorrosion test on the TiN film showed lines of adhered material in the worn zone, which were caused by debris accumulation that contributed to the abrasive marks on the coated surfaces. The wear track on the ball used for the TiCN tribocorrosion tests had adhered material at the borders and a tribolayer in the center, which could lead to further abrasion marks on the worn zones of the coated surfaces. In contrast, the worn zone of the counter body tested with the TiCN film exhibited a soft surface with material accumulation at the worn zone borders and some abrasion marks in the center of the wear track.

#### 3.4.2. Friction Force

Figure 7a shows the friction force (Ff) observed during tribocorrosion tests conducted on D2 steel surfaces coated with TiN, TiCN, and TiCrN. The Ff exhibited two distinct phases during the tests. In the initial phase, the friction force increased from 0.1 N to a maximum of 0.43 N, then decreased to 0.3 N. This fluctuation in Ff at the start of the tribocorrosion tests was attributed to surface adaptation involving the plastic deformation and fracturing of asperities and surface alterations caused by the corrosion process during the passivation period of the D2 steel surfaces. The second phase was more stable, with a coefficient of friction (CoF) value of 0.28 ± 0.02. For the TiN film, the Ff in the initial phase stabilized around 0.9 N before increasing to yield a CoF of 0.16 ± 0.01. The change in Ff was due to the removal of products from the passivation period. The tribocorrosion performance of the TiCN film was similar to that of the D2 steel surfaces, beginning with an initial Ff of 0.11 N, which rose to a maximum of 0.22 N before stabilizing at a CoF value of 0.18 ± 0.01. In contrast, the Ff values for the TiCrN film showed an increment from 0.15 N, reaching up to 0.7 N, resulting in a stable CoF value of 0.69 ± 0.02 (see Figure 8). This CoF value is atypical for tribocorrosion tests, where the corrosive solution generally acts as a lubricant, as seen with the D2 steel, TiN, and TiCN films. However, in the case of the TiCrN film, the elevated CoF value was attributed to more significant surface deformation of the layer and enhanced adhesion between the corrosion products in the worn zone and the ZrO_2_ ball. This behavior was evident on the wear track of the TiCrN film, which displayed abrasive marks and plastic deformation of the tribolayer during the sliding tests [88]. Similar CoF value behaviors were reported by Chen et al. [18], who noted fracture and delamination failures in the worn zones of wear tracks during tribocorrosion testing.

#### 3.4.3. Open Circuit Potential (OCP)

Figure 7b presents the open circuit potential (OCP) measured during tribocorrosion tests on D2 steel coated with TiN, TiCN, and TiCrN films. Initially, the OCP on the D2 steel surfaces increases at the beginning of the rubbing operation and remains relatively constant during the sliding test. This increase in OCP is attributed to a reduction in porosity of the film and the porosity caused by the corrosion process and the plastic deformation of the surface, which minimizes the exposed area to the corrosive environment [99,100,101]. For the TiN film, the OCP value decreases at the start of the sliding operation due to the wear of the passive layer on the surface and ongoing oxidation processes throughout the tests. The OCP measured on the TiCN film follows a similar trend to the TiN film; however, the reduction in OCP on the TiCN film is less pronounced, indicating better stability during the tribocorrosion tests. In the case of the TiCrN film, the OCP increases at the beginning of the rubbing operation. This increase is due to a decrease in the exposed area to the corrosion solution, resulting from the plastic deformation of the film material, which helps cover the porosity and delamination of the worn surfaces. The OCP value produced on the TiN film decreased at the start of the sliding operation due to the wear of the passive layer on the surface and the oxidation process of the exposed surface that continued during the tests. The OCP value measured during the tribocorrosion test on the TiCN film had a similar performance to the OCP on the TiN film, with the difference that the reduction in the OCP value on the TiCN film was smaller than on the TiN film, showing better stability during the tribocorrosion tests. The OCP produced during the tribocorrosion test on TiCrN increased at the start of the rubbing operation due to the reduction in the exposed area to the corrosion solution by the plastic deformation of the film material, covering the porosity and delamination of the worn surfaces [51,101,102].

## 4. Discussion

The wear caused by corrosion and tribocorrosion significantly impacts the efficiency and lifetime of cutting tools in the food industry, reducing their edge sharpness and overall durability. The cutting tools commonly used in kitchens and restaurants are made from stainless steel. However, in the food industry, utilizing materials that are more resistant to plastic deformation, corrosion, and tribocorrosion is essential, especially in corrosive environments, such as those encountered with processing fish, meat, and vegetables [2,5,6]. D2 steel is employed in various industrial applications due to its favorable mechanical and corrosion properties [9,11]. However, it exhibits low resistance to wear in corrosive environments. To enhance the tribocorrosion performance of D2 steel, covering its surfaces with a protective film layer could minimize exposure to corrosive elements. The D2 steel surfaces in this study showed standard elemental composition and hardness values. Electrochemical tests revealed pitting with a granular morphology, indicating corrosion effects that accelerated the corrosion process and resulted in the increment of the corrosion current (Icorr). These corrosion effects were also observed during tribocorrosion testing, albeit in a different manner. The plastic deformation produced during ribbing operation on the steel surfaces reduced the impact of the pitting, as it diminished the exposed area to the corrosive environment and affected the open circuit potential (OCP) [99,100]. Because of their excellent mechanical and tribological properties, titanium nitride (TiN) films have been studied for various applications, including cutting tools and biomedical components. In this work, the TiN film exhibited a sodium chloride (NaCl) crystalline structure with an elemental composition of 70 atom% titanium (Ti) and 30 atom% of nitrogen (N), indicating that the layer consisted of both TiN and metallic Ti phases. These characteristics resulted in a TiN layer with hardness values lower than previously reported and high plastic deformation during the scratch tests but with similar electrochemical and tribocorrosion resistance [31,33,103,104]. On the D2 steel, the TiN layer significantly improved the mechanical, electrochemical, and tribocorrosion properties (see Figure 8).

Carbon (C) and chromium (Cr) were included during the deposition process to further enhance the properties of the TiN layer. The inclusion of carbon produced a titanium carbonitride (TiCN) layer with a composition of 38 atm% of Ti, 45 atm% of N, and 17 atm% of C, which contained TiN and titanium carbide (TiC) crystalline phases, along with a carbon-carbon bond in the graphitic phase. Another phase, a-cyanide (aCNx), could potentially form during deposition; however, X-ray diffraction (XRD) and Raman spectroscopy did not provide evidence of its presence. The TiCN layer exhibited comparable hardness and adhesion to the TiN film but demonstrated lower electrochemical properties, shown by a higher Icorr value and reduced protection percentage (Pi%). Although the TiCN film had a lower OCP value during tribocorrosion testing than the TiN film, TiCN exhibited better stability, with a smaller reduction in the OCP value during the rubbing operation. The stability of the OCP during tribocorrosion can be attributed to the carbon phases within the layer, which enhance corrosion resistance stability during rubbing operations [34,105,106,107]. Including chromium during the TiN film deposition resulted in a titanium chromium nitride (TiCrN) layer with a composition of 34 atom% of Ti, 30 atom% of N, and 36 atom% of Cr. The TiCrN layer included TiN, chromium nitride (CrN), and Cr phases, showing a higher intensity peak corresponding to the (220) CrN phase and (200) Cr metallic phase. This modification indicates a transition from the primary TiN crystalline structure to the CrN and Cr crystalline structures within the TiCrN layer. While the TiCrN film presented higher hardness than the TiN film, it exhibited inferior adhesion and electrochemical and tribocorrosion performance, marked by spallation of the layer during the scratch test and the highest Icorr, a negative polarization resistance, and a greater wear rate and coefficient of friction (CoF) (see Figure 8). A similar CoF value was reported by Chen et al. [18], with the difference that the OCP registered during the rubbing process decreased. The TiCrN film’s OCP performance was similar to that of the D2 substrate surface, with an increase in OCP during the rubbing operation, suggesting that plastic deformation of the surfaces and the formation of a tribolayer (Tl) mitigated the effects of pitting and porosity during the wear process. The increment in the macroparticles deposition with the addition of C and the increased porosity with the addition of Cr to TiN film is a factor in the reduction in the electrochemical and wear properties of the TiN, producing a greater number of cracks and increased porosity during the cooling process, that increased the corrosive solution filtration in the films, especially for the TiCrN film. In the same way, the macroparticles presented lower cohesion to the TiN matrix, increasing the wear rate due to these particles, working like an abrasive body in the rubbing operation [49,101,102]. The effects on the mechanical, corrosion, tribological, and tribocorrosion properties and the formation of the macroparticles produced during the deposition process are not completely understood. Some authors, such as Muhammed et al. [28], reported that the macroparticles could be formed by thermal shock, hydrodynamic effects, or micro-explosion on the cathode caused by high current during the erosion process that, combined with the deposition parameters, adds Mps with different size, shape, density, and fly velocity to the substrate. The inclusion of Mps in the film during the deposition process could modify the film characteristics, such as the thickness, cohesion, homogeneity, roughness, and elemental composition, modifying the stress distribution, elastoplastic performance, fatigue resistance, and chemical stability, among other properties that reduced the efficiency of the films [108,109]. Some projects have been dedicated to studying techniques to reduce the Mps in the film produced by the catholic arc technique. Baseri et al. [110] reported that increment in the bias voltage reduced Mp production and porosity in the layer, improving the corrosion resistance of the CrN/CrAlN films. Adhesina et al. [111], for example, reported reduction in the porosity percentage in CrAlN and TiAlN with the use and increase of the bias voltage, improving the corrosion resistance of the films. In this work, a bias of 250 V was used for each target, reducing the Mps on the TiN film; however, the bias voltages presented a lower reduction in the Mps during the deposition of TiN films with C or Cr that may have been produced by changes in the variation in the target surfaces (poisoning of the surfaces targets) and its effect on the porosity and defects in the film matrix and the macroparticles in and on the coated surfaces. In the fishing, meat, vegetable, and other food industries, the environment presents a corrosive atmosphere, and improvement in the corrosion and tribocorrosion resistance of the cutting tools is necessary to improve food processing efficiency.

## 5. Conclusions

The deposition of titanium nitride (TiN) using the PVD-cathodic arc technique on D2 steel surfaces enhances their mechanical, electrochemical, and tribological properties, making TiN a suitable choice for cutting tools in the food industry. The presence of carbon (C) and chromium (Cr) during the deposition of the TiN film negatively affected its properties. These elements led to increased porosity and defects in the films and macroparticles. As a result, the film’s surface area exposed to the corrosive solution increased, which enhanced both the mechanical and corrosion wear rates. Using these cathodic arc parameters negatively affected the properties of the TiN film, particularly its corrosion and tribocorrosion protection. To improve these properties, introducing a ternary element into the TiN matrix could be beneficial, but it is crucial that the element is incorporated using the appropriate deposition parameters to enhance protective qualities.

## Figures and Tables

**Figure 1 materials-18-02733-f001:**
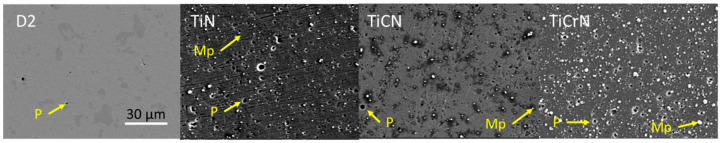
Surface morphology of the D2 steel uncoated and coated with TiN, TiCN, and TiCrN films.

**Figure 2 materials-18-02733-f002:**
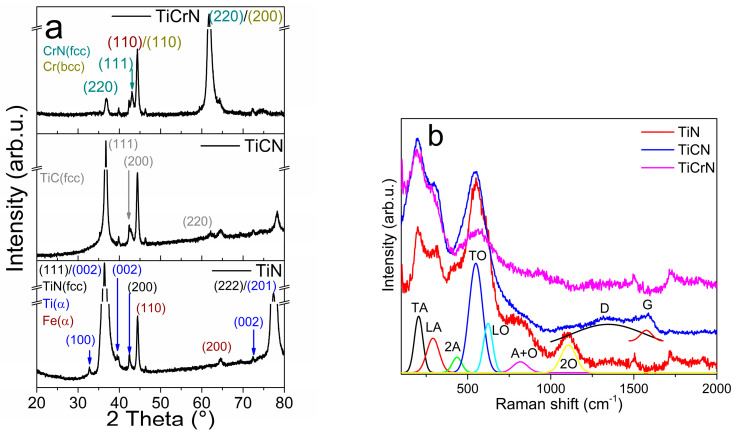
(**a**) XRD patterns and (**b**) Raman spectra of the D2 steel surfaces coated with TiN, TiCN, and TiCrN films.

**Figure 3 materials-18-02733-f003:**
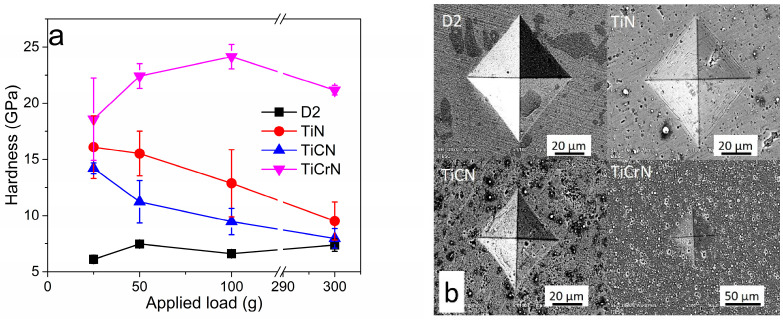
(**a**) Hardness and (**b**) indentation mark of D2 steel coated with TiN, TiCN, and TiCrN films.

**Figure 4 materials-18-02733-f004:**
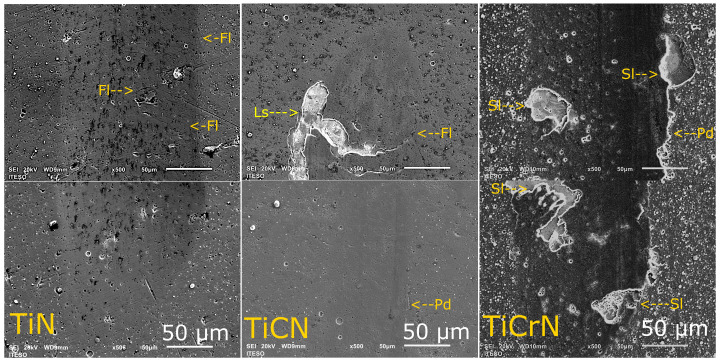
Scratch test images of TiN, TiCN, and TiCrN films (yellow; Fl = fracture layer, Ls = layer spallation, Pd = plastic deformation).

**Figure 5 materials-18-02733-f005:**
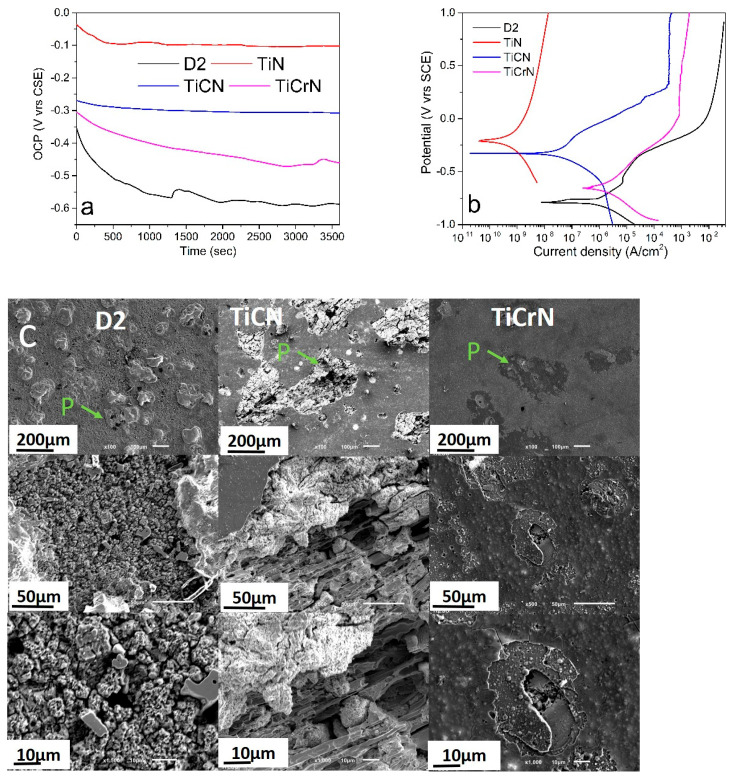
(**a**) OCP, (**b**) Potentiodynamic polarization curves (Tafel curves) and (**c**) corrosion wear marks produced during the electrochemical testing of D2 steel coated with TiN, TiCN, and TiCrN films (P = Pitting).

**Figure 6 materials-18-02733-f006:**
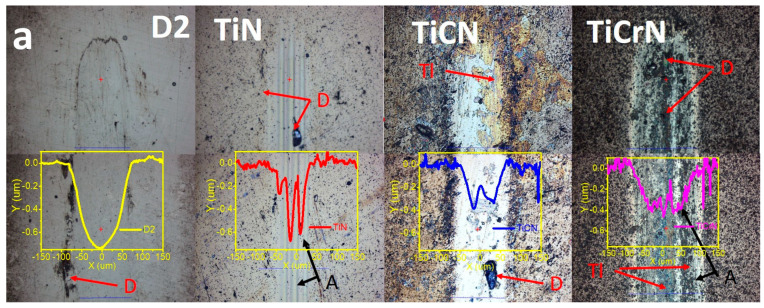
(**a**) Wear rate and wear profiles of the D2 steel coated with TIN, TiCN, and TiCrN films produced for the tribocorrosion tests (yellow = wear track profile, red color = wear mechanism, D = debris, Tl = tribolayer and A = abrasion) and (**b**) wear track on the pin surfaces (M = material transferred).

**Figure 7 materials-18-02733-f007:**
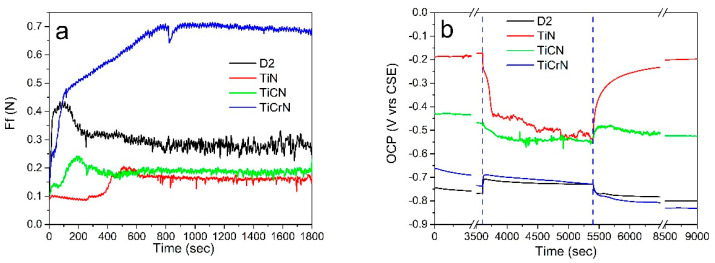
(**a**) Friction force and (**b**) OCP produced during the tribocorrosion tests on D2 steel surfaces coated with TiN, TiCN, and TiCrN films.

**Figure 8 materials-18-02733-f008:**
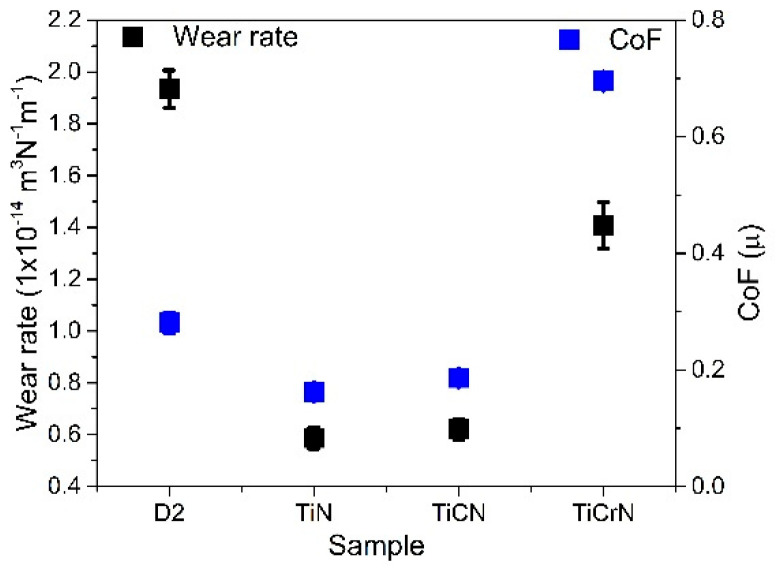
Wear rate and CoF value produced during the tribocorrosion tests on D2 steel surfaces coated with TiN, TiCN, and TiCrN films.

**Table 1 materials-18-02733-t001:** Electrochemical properties of D2 steel surfaces coated with TiN, TiCN, and TiCrN films.

Surfaces	Ecorr (mV)	Icorr (μA/cm^2^)	βa (mV/dec)	βc (mV/dec)	Rp (kΩ/cm^2^)	Pi%
D2	−791.7	3.66 × 10^−1^	90.0	−45.0	35.1	
TiN	−212.5	1.17 × 10^−4^	45.0	−85.0	109,890.1	99.97
TiCN	−328.9	1.16 × 10^−2^	90.0	−44.0	1103.1	96.82
TiCrN	−655.2	5.17 × 10^−1^	70.0	−52.0	24.8	−41.23

## Data Availability

The data will be available on request.

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
