# Peer review of "Electrochemical and Tribocorrosion Study of D2 Steel Coated with TiN with C or Cr Addition Films in 3.5 wt% of NaCl in Bi-Distillated Water Solution"

_materials, 2025, doi:10.3390/ma18122733_

Round 1
Reviewer 1 Report
Comments and Suggestions for Authors
Reviewer comments
The manuscript deals with an important topic of “Electrochemical and tribocorrosion study of D2 steel coated with TiN with C or Cr addition films in 3.5 wt% of NaCl in bi-distillated water solution”. This work presents the electrochemical and tribocorrosion studies of D2 steel surfaces coated with TiN, TiCN and TiCrN films. The samples were coated by a commer-cial source, using the PVD-Cathodic arc technique. The crystalline structure TiN and TiCN films presented TiN and Ti phase, while the crystalline structure of TiCrN shows the CrN and Cr phases. However, certain limitations exist that, if addressed, could enhance the comprehensiveness and applicability of the findings. It is suggested to be accepted after major revision. The specific comments are the following.
- Clarify the Significance of the Study in the Abstract: The abstract currently lacks a clear statement of the study's broader significance beyond the technical details. I recommend emphasizing how your findings will impact the food industry and the potential applications of TiN, TiCN, and TiCrN coatings in improving the longevity and efficiency of cutting tools. Strengthening the "so what" aspect will help readers understand the practical value of your work.
Revision Suggestion: Add a sentence at the end of the abstract stating the broader implications of your research on improving food processing efficiency and extending the service life of cutting tools in challenging environments.
- Improve the Clarity and Flow of the Introduction: The introduction could benefit from a clearer structure and flow. It currently jumps between various aspects of the study without adequately connecting the dots for the reader.
Revision Suggestion: Revise the introduction to first provide background on the challenges of cutting tools in the food industry, then introduce D2 steel and coatings, and finally, position your study as a response to these challenges. This will guide the reader more smoothly through the argument.
- Expand on the Role of Carbon and Chromium in Coatings: In several sections, the manuscript touches upon the impact of carbon and chromium in TiN coatings but does not elaborate sufficiently on why these elements were chosen or how they interact with the titanium matrix. Given their significant role, this should be better explained.
Revision Suggestion: Include more detailed explanations on the mechanisms by which carbon and chromium enhance the coatings' properties, especially in terms of tribocorrosion resistance and wear resistance.
- The section of introduction is weak. For the part of “Introduction”, the lack of literature references lowers the credibility and scientific nature of the research basis. A large sum of references is necessary to be added. It is suggested to establish a quantitative relationship between preparation technology and design development. Compared with other works, such as Journal of Materials Research and Technology 2024 (https://doi.org/10.1016/j.jmrt.2024.11.024).
- Provide More Data on Comparative Performance: In the Results section, the performance of the coatings (TiN, TiCN, and TiCrN) is presented in isolation without enough comparative analysis. Although some comparisons are made, they lack statistical backing.
Revision Suggestion: Add comparative data (e.g., statistical analysis like ANOVA) between the different coatings to reinforce the validity of your claims regarding their performance. This would strengthen the argument that one coating significantly outperforms the others.The hybrid approach (trochoidal roughing followed by conventional finishing) is briefly mentioned, but it could benefit from further elaboration. A clearer analysis of the specific advantages of this approach in terms of tool wear, cycle times, surface quality, and cost-efficiency should be included.
- Refine the Discussion on Tribocorrosion Behavior: The discussion on the tribocorrosion results could be more nuanced. While it mentions the behavior of the coatings, it does not sufficiently explain why TiCrN, despite higher wear resistance, performed worse in terms of electrochemical properties.
Revision Suggestion: Deepen the analysis of TiCrN's tribocorrosion behavior, specifically addressing how the increased macroparticles and porosity might have affected the electrochemical behavior and why this resulted in a reduced protection percentage.
- Enhance Figure Legends for Clarity: Some of the figure legends could benefit from being more descriptive. For example, the legend for Figure 5 describing the electrochemical properties doesn't explain the key differences in behavior between the coatings.
Revision Suggestion: Revise the figure legends to include more context. For example, specify what each curve represents and explain what the reader should take away from each figure, especially highlighting the differences between the coatings.
- Consolidate the Results and Discussion Sections: In its current form, the Results section is slightly fragmented with lots of technical details, and the Discussion section seems to repeat some of these findings. A more integrated approach could improve the readability and make the findings easier to follow.
Revision Suggestion: Consider integrating the Results and Discussion sections so that each result is immediately followed by its interpretation. This would make the paper more fluid and easier to digest for readers.
- Strengthen the Conclusion with Future Directions: The conclusion briefly touches on potential future research but doesn't outline specific avenues for further investigation.
Revision Suggestion: Expand the conclusion by suggesting specific future research topics, such as the exploration of other ternary elements or the use of other deposition techniques to improve coating properties.
- Improve Consistency in Terminology: There are some inconsistencies in terminology, especially in the materials and methods section where "film" and "coating" are used interchangeably. It may confuse readers who expect consistency throughout.
Revision Suggestion: Standardize the terminology to consistently use either "film" or "coating" throughout the manuscript to improve clarity.
- The language description of the article needs to be standardized. Please check it carefully and avoid using unprofessional words and sentences.
Comments on the Quality of English Language
The English could be improved.
Author Response
The manuscript deals with an important topic of “Electrochemical and tribocorrosion study of D2 steel coated with TiN with C or Cr addition films in 3.5 wt% of NaCl in bi-distillated water solution”. This work presents the electrochemical and tribocorrosion studies of D2 steel surfaces coated with TiN, TiCN and TiCrN films. The samples were coated by a commer-cial source, using the PVD-Cathodic arc technique. The crystalline structure TiN and TiCN films presented TiN and Ti phase, while the crystalline structure of TiCrN shows the CrN and Cr phases. However, certain limitations exist that, if addressed, could enhance the comprehensiveness and applicability of the findings. It is suggested to be accepted after major revision. The specific comments are the following.
- Clarify the Significance of the Study in the Abstract: The abstract currently lacks a clear statement of the study's broader significance beyond the technical details. I recommend emphasizing how your findings will impact the food industry and the potential applications of TiN, TiCN, and TiCrN coatings in improving the longevity and efficiency of cutting tools. Strengthening the "so what" aspect will help readers understand the practical value of your work.
Revision Suggestion: Add a sentence at the end of the abstract stating the broader implications of your research on improving food processing efficiency and extending the service life of cutting tools in challenging environments.
The next text was added to the manuscript.
Due to the cutting tools being exposed to a high-demand environment that includes high contact pressure, corrosive atmosphere, and high-speed process, these presented a high mechanical and corrosive wear that reduced their lifetime and efficiency. Tribocorrosion is one of the main phenomena that reduces the lifetime and efficiency of the cutting tool.
- Improve the Clarity and Flow of the Introduction: The introduction could benefit from a clearer structure and flow. It currently jumps between various aspects of the study without adequately connecting the dots for the reader.
Revision Suggestion: Revise the introduction to first provide background on the challenges of cutting tools in the food industry, then introduce D2 steel and coatings, and finally, position your study as a response to these challenges. This will guide the reader more smoothly through the argument.
The nest text was added to the manuscript:
In the same the corrosion is one of the most important issue that reduces the life-time of the tools and infrastructure, using stainless steel and aluminum alloy for the most of the metallic element in the food industry [7, 8]. However, there are operation that require a high wear and stress resistance such as cutting operation. One example is the Fisheries and aquaculture industry that are among the most significant sectors in the food industry [5], vegetable and fruits processing [9, 10], meat [11] and other food sectors.
For that, the modification of the AISI D2 steel surfaces in order to improve corrosion, wear and tribocorrsion properties has been studied in several projects, some of these are; Kai-gude et al. [24] reported the use of the Electrical Discharge Machining (EDM) to use like a matching process; Reséndiz-Calderón et al. [25]reported the increment of wear properties of the D2 steel due to the application of a boriding thermal process; Castillejo et al. [26] re-ported the increment of the wear and corrosion property of the D2 steel using the ther-mos-reactive process to deposit a Chromium–Vanadium Carbide Films; Voglar et al. [27] used a cryogenic treatment to modify the hardness and corrosion properties of the D2 steel surfaces.
- Expand on the Role of Carbon and Chromium in Coatings: In several sections, the manuscript touches upon the impact of carbon and chromium in TiN coatings but does not elaborate sufficiently on why these elements were chosen or how they interact with the titanium matrix. Given their significant role, this should be better explained.
Revision Suggestion: Include more detailed explanations on the mechanisms by which carbon and chromium enhance the coatings' properties, especially in terms of tribocorrosion resistance and wear resistance.
- The section of introduction is weak. For the part of “Introduction”, the lack of literature references lowers the credibility and scientific nature of the research basis. A large sum of references is necessary to be added. It is suggested to establish a quantitative relationship between preparation technology and design development. Compared with other works, such as Journal of Materials Research and Technology 2024 (https://doi.org/10.1016/j.jmrt.2024.11.024).
Several references were included to the introduction section in order to improve the quantitative relationship between preparation technology and design development.
- Provide More Data on Comparative Performance: In the Results section, the performance of the coatings (TiN, TiCN, and TiCrN) is presented in isolation without enough comparative analysis. Although some comparisons are made, they lack statistical backing.
Revision Suggestion: Add comparative data (e.g., statistical analysis like ANOVA) between the different coatings to reinforce the validity of your claims regarding their performance. This would strengthen the argument that one coating significantly outperforms the others.The hybrid approach (trochoidal roughing followed by conventional finishing) is briefly mentioned, but it could benefit from further elaboration. A clearer analysis of the specific advantages of this approach in terms of tool wear, cycle times, surface quality, and cost-efficiency should be included.
The next texts were included in “result” section in order to improve the comparative performance of the films.
Mechanical properties section:
In Figure 1 shows that the inclusion of C and Cr in the TiN film increased the roughness due to the increment of marcroparticles and porosity on the layers, having a higher defects the TiN and TiCrN films. The increase in porosity and macroparticles in TiN, due to the addition of carbon (C) and chromium (Cr), can be attributed to a chemical reaction be-tween the metallic target (titanium, Ti) and the reactive atmosphere containing nitrogen (N) and carbon. This reaction leads to the formation of titanium carbide (TiC) and titani-um nitride (TiN) on the surface of the targets in the deposition of TiCN films or TiN and CrN in the deposition of TiCrN films increasing the melting point of the surfaces and spallation of macroparticles during the evaporation process, which then deposit onto the substrate [45-47].
Mechanical properties section:
Although, the reported value of the harness of the TiN films are from 13.3 GPa [60], and variety with the film thickness [61], the lower hardness value of the TiN and TiCN films than the reported could be produced by the substrate effect due to effects of the plastic de-formation of the substrate, causing that the hardness measurement were a composite hardness [62].
Electrochemical section:
Although some works have been reported the increment of the corrosion resistance of bi-nary layer (metallic-carbide MC or metallic-nitride MN films) with the addition a ternary material such as Cr and C, the defects and morphology of the films can reduced this per-formance [77-79]. The addition of carbon (C) and chromium (Cr) has been utilized to en-hance the mechanical and electrochemical properties of materials, particularly due to Cr. The inclusion of chromium can lead to the formation of a Cr2O3 film, which exhibits high chemical and mechanical stability [45, 80-82]. However, these findings indicate a decrease in the corrosion resistance of the TiN film when C and Cr are added. This reduction can be attributed to the increased filtration of corrosive solutions through the porosity and defects present in the TiN matrix, which are caused by macroparticles and variations in the composition generated during the deposition process [73, 81-83]. In the same way, the porosity and defects of the TiCN and TiCrN films increased the exposed area to the corro-sive solution, increasing the current density (icorr). This effect can be observed in the vari-ation of the Pi, where the TiCN and TiCrN films presented a lower Pi value than the TiN film, especially the Pi value of the TiCrN film, where the increment of the area improves the Redox process, causing a higher ion liberation that produced a higher icorr on the coated D2 steel surfaces with TiCrN film than the Uncoated D2 surfaces.
Discussion section
The effects in the mechanical, corrosion, tribological and tribocorrosion properties and the formation of the macroparticles produced during the deposition process are not complete understanding. Some authors such as Muhammed et al. [28] reported that the macropar-ticle could be formed by thermal shock, hydrodynamic effects, micro-explosion on the cathode caused by high current during the erosion process that combined with the depo-sition parameters produce Mp with different size, shape, density and fly velocity to the substrate. The inclusion of Mp in the film during the deposition process could modifies the film characteristics such as thickness, cohesion, homogeneity, roughness and ele-mental composition, modifying the stress distribution, elastoplastic performance, fatigue resistance, chemical stability, between other properties that reduced the film efficiency of the films [93, 94]. For that, some projects have been dedicated to study some techniques to reduce the Mp in the film produced by catholic arc technique. Baseri et al. [95] reported that the increment of the bias voltage reduced the Mp production and porosity in the layer, improving the corrosion resistance of CrN/CrAlN films; Adhesina et al. [96] reported the reduction of porosity percentage in CrAlN and TiAlN, improving the corrosion resistance of the films. For that, in this work was used a bias of 250 V for each target, reducing the Mp on the TiN film, however the bias voltages presented a lower reduction of the Mp during the deposition of TiN films with C or Cr. For that in the fishing, meat, vegetable and other food industries the environment presents a corrosive atmosphere.
- Refine the Discussion on Tribocorrosion Behavior: The discussion on the tribocorrosion results could be more nuanced. While it mentions the behavior of the coatings, it does not sufficiently explain why TiCrN, despite higher wear resistance, performed worse in terms of electrochemical properties.
Revision Suggestion: Deepen the analysis of TiCrN's tribocorrosion behavior, specifically addressing how the increased macroparticles and porosity might have affected the electrochemical behavior and why this resulted in a reduced protection percentage.
The effects in the mechanical, corrosion, tribological and tribocorrosion properties and the formation of the macroparticles produced during the deposition process are not complete understanding. Some authors such as Muhammed et al. [28] reported that the macropar-ticle could be formed by thermal shock, hydrodynamic effects, micro-explosion on the cathode caused by high current during the erosion process that combined with the depo-sition parameters produce Mp with different size, shape, density and fly velocity to the substrate. The inclusion of Mp in the film during the deposition process could modifies the film characteristics such as thickness, cohesion, homogeneity, roughness and ele-mental composition, modifying the stress distribution, elastoplastic performance, fatigue resistance, chemical stability, between other properties that reduced the efficiency of the films [93, 94]. For that, some projects have been dedicated to study some techniques to re-duce the Mp in the film produced by catholic arc technique. Baseri et al. [95] reported that the increment of the bias voltage reduced the Mp production and porosity in the layer, improving the corrosion resistance of CrN/CrAlN films; Adhesina et al. [96] reported the reduction of porosity percentage in CrAlN and TiAlN with the use and increment of the bias voltage, improving the corrosion resistance of the films. For that, in this work was used a bias of 250 V for each target, reducing the Mp on the TiN film, however the bias voltages presented a lower reduction of the Mp during the deposition of TiN films with C or Cr that can be produced by the changes in the variation of the target surfaces (poisoning of the surfaces targets) and its effect in the porosity and defects in the film matrix and the macroparticles in and on the coated surfaces. For that in the fishing, meat, vegetable and other food industries the environment presents a corrosive atmosphere, the improvement of the corrosion and tribocorrosion resistance of the cutting tools are necessary to improve the food processing efficiency.
- Enhance Figure Legends for Clarity: Some of the figure legends could benefit from being more descriptive. For example, the legend for Figure 5 describing the electrochemical properties doesn't explain the key differences in behavior between the coatings.
Revision Suggestion: Revise the figure legends to include more context. For example, specify what each curve represents and explain what the reader should take away from each figure, especially highlighting the differences between the coatings.
The next text was included in the manuscript
These effects can be observed in the SEM images of the corrosion were on the D2 steel sur-faces uncoated and coated with TiCN and TiCrN films (see Figure 5c). In this was ob-served that the corrosion wear marks on D2 steel presented a granular morphology that were produced by the corrosion process on the grain borders. The corrosion marks ob-served in the TiCN film displayed a columnar morphology, which is likely produced by the filtration of material within the TiN matrix along the edges of the film's columnar structure. Similarly, the corrosion marks in the TiCrN film indicate a comparable filtration process occurring through its columnar morphology. However, the corrosion marks on the TiCrN film also exhibit delamination and spallation in certain layers, suggesting that the filtration occurred in both columnar and transversal directions.
- Consolidate the Results and Discussion Sections: In its current form, the Results section is slightly fragmented with lots of technical details, and the Discussion section seems to repeat some of these findings. A more integrated approach could improve the readability and make the findings easier to follow.
Revision Suggestion: Consider integrating the Results and Discussion sections so that each result is immediately followed by its interpretation. This would make the paper more fluid and easier to digest for readers.
- Strengthen the Conclusion with Future Directions: The conclusion briefly touches on potential future research but doesn't outline specific avenues for further investigation.
Revision Suggestion: Expand the conclusion by suggesting specific future research topics, such as the exploration of other ternary elements or the use of other deposition techniques to improve coating properties.
The next text was included in the manuscript:
The effects in the mechanical, corrosion, tribological and tribocorrosion properties and the formation of the macroparticles produced during the deposition process are not com-plete understanding. Some authors such as Muhammed et al. [28] reported that the mac-roparticle could be formed by thermal shock, hydrodynamic effects, micro-explosion on the cathode caused by high current during the erosion process that combined with the deposition parameters produce Mp with different size, shape, density and fly velocity to the substrate. The inclusion of Mp in the film during the deposition process could modi-fies the film characteristics such as thickness, cohesion, homogeneity, roughness and ele-mental composition, modifying the stress distribution, elastoplastic performance, fatigue resistance, chemical stability, between other properties that reduced the film efficiency of the films [93, 94]. For that, some projects have been dedicated to study some techniques to reduce the Mp in the film produced by catholic arc technique. Baseri et al. [95] reported that the increment of the bias voltage reduced the Mp production and porosity in the layer, improving the corrosion resistance of CrN/CrAlN films; Adhesina et al. [96] reported the reduction of porosity percentage in CrAlN and TiAlN, improving the corrosion resistance of the films. For that, in this work was used a bias of 250 V for each target, reducing the Mp on the TiN film, however the bias voltages presented a lower reduction of the Mp during the deposition of TiN films with C or Cr. For that in the fishing, meat, vegetable and other food industries the environment presents a corrosive atmosphere.
- Improve Consistency in Terminology: There are some inconsistencies in terminology, especially in the materials and methods section where "film" and "coating" are used interchangeably. It may confuse readers who expect consistency throughout.
Revision Suggestion: Standardize the terminology to consistently use either "film" or "coating" throughout the manuscript to improve clarity.
The “film” term was used in the manuscript
- The language description of the article needs to be standardized. Please check it carefully and avoid using unprofessional words and sentences
The text in the manuscript was revised in order to avoid unprofessional words and sentences.

Reviewer 2 Report
Comments and Suggestions for Authors
The authors can found and download the reviewer's report in PDF document below

Comments on the Quality of English Language
English should be refined
Author Response
Answer to the Review Report
Manuscript Title: Electrochemical and Tribocorrosion Study of D2 Steel Coated with TiN with C or Cr Addition Films in 3.5 wt% of NaCl in Bidistillated Water Solution.
Manuscript ID: materials-3525000
This manuscript presents a detailed study into the electrochemical and tribocorrosion properties of D2 steel substrates coated with TiN, TiCN, and TiCrN films, prepared via the PVD-Cathodic Arc technique. The study aims to contribute to the improvement of cutting tool performance in the food industry, where tribocorrosion is a significant challenge. The work combines a variety
of experimental techniques to characterize the coatings’ structural, mechanical,
electrochemical, and tribological properties.
The research demonstrates some originality in exploring ternary TiN-based coatings (TiCN and TiCrN) on D2 steel under conditions relevant to food processing (3.5 wt% NaCl solution).
However, the manuscript has several shortcomings that affect its overall quality.
The following are the reviewer’s questions and comments to be addressed by the authors:
- The authors should expand the introduction to highlight gaps in prior studies (e.g., specific to D2 steel or food industry conditions) and explicitly state the novel aspects of this work. The novelty appears incremental rather than transformative, and a stronger case for originality is needed. Additionally, it was noticed that many references are out of date. The authors are encouraged to include fresh reported literature at least in the last five years.
In the work was included the next text:
For that, the modification of the AISI D2 steel surfaces in order to improve corrosion, wear and tribocorrsion properties has been studied in several projects, some of these are; Kai-gude et al. [19] reported the use of the Electrical Discharge Machining (EDM) to use like a matching process; Reséndiz-Calderón et al. [20]reported the increment of wear properties of the D2 steel due to the application of a boriding thermal process; Castillejo et al. [21] re-ported the increment of the wear and corrosion property of the D2 steel using the ther-mos-reactive process to deposit a Chromium–Vanadium Carbide Coatings; Voglar et al. [22] used a cryogenic treatment to modify the hardness and corrosion properties of the D2 steel surfaces.
- Related to the literature review, the introduction provides a broad context for the study, linking food security, cutting tool performance, and tribocorrosion. However, the literature review is incomplete and lacks depth in key areas: for instance, the discussion of TiN, TiCN, and TiCrN cites foundational studies, but omits recent advancements or controversies, such as the role of macroparticles in PVD coatings or the trade-offs between hardness and corrosion resistance in ternary coatings. Moreover, fisheries applications are mentioned, the review does not connect prior coating studies to food processing environments, limiting its relevance.
In the work, the next text was included:
Introduction section
In the same the corrosion is one of the most important issue that reduces the life-time of the tools and infrastructure, using stainless steel and aluminum alloy for the most of the metallic element in the food industry [7, 8]. However, there are operation that require a high wear and stress resistance such as cutting operation. One example is the Fisheries and aquaculture industry that are among the most significant sectors in the food industry [5], vegetable and fruits processing [9, 10], meat [11] and other food sectors.
Discussion section
The effects in the mechanical, corrosion, tribological and tribocorrosion properties and the formation of the macroparticles produced during the deposition process are not com-plete understanding. Some authors such as Muhammed et al. [28] reported that the mac-roparticle could be formed by thermal shock, hydrodynamic effects, micro-explosion on the cathode caused by high current during the erosion process that combined with the deposition parameters produce Mp with different size, shape, density and fly velocity to the substrate. The inclusion of Mp in the coating during the deposition process could modifies the coating characteristics such as thickness, cohesion, homogeneity, roughness and elemental composition, modifying the stress distribution, elastoplastic performance, fatigue resistance, chemical stability, between other properties that reduced the coating ef-ficiency of the coatings [77, 78]. For that, some projects have been dedicated to study some techniques to reduce the Mp in the coating produced by catholic arc technique. Baseri et al. [79] reported that the increment of the bias voltage reduced the Mp production and poros-ity in the layer, improving the corrosion resistance of CrN/CrAlN coatings; Adhesina et al. [80] reported the reduction of porosity percentage in CrAlN and TiAlN, improving the corrosion resistance of the films. For that, in this work was used a bias of 250 V for each target, reducing the Mp on the TiN film, however the bias voltages presented a lower re-duction of the Mp during the deposition of TiN films with C or Cr. For that in the fishing, meat, vegetable and other food industries the environment presents a corrosive atmos-phere.
- The PVD-Cathodic Arc technique is described as a “commercial source”, but critical parameters (e.g., arc current, gas pressure, substrate temperature) are not provided, hindering replication. The use of a 3.5 wt% NaCl solution and potentiodynamic testing is relevant to the study’s aims. However, the 1-hour passivation period, as stated in the manuscript, lacks justification, and triplicate testing or error analysis is not mentioned.
Experimental section:
The TiN coating containing carbon and chromium was deposited using the PVD-Cathodic Arc technique from a commercial source. Ti and Cr targets, each with a purity of 99.99%, were utilized along with N2 and C2H2 gases as precursors for producing nitride, carbide, and carbonitride films. To produce the TiN coatings, four titanium targets were employed, operating at 85 A in an N2 atmosphere with a working pressure of 3.5 Pa and a bias voltage of 250 V. For the TiCN coating, the same parameters used for the TiN film were applied, with the addition of 10% C2H2 in the N2 gas mixture. For the TiCrN coating, the same parameters as those for the TiN coating were used, but with three chromium targets, also operating at 85 A.….
Prior to the potentiodynamic tests, a 1-hour passivation period was observed (This stand-ard time was determined from the stabilization time of OCP voltages for the samples).
- The choice of 1 N load and 0.5-hour duration in the reciprocating sliding testing is arbitrary without reference to food industry conditions or prior standards. Wear track analysis uses optical profilometry, but quantification methods (e.g., wear volume calculation) are unclear.
This text was added to the work:
These tests parameter were selected to overcome the cutting operation in the food industry due to these caused a Hertzian contact pressure of 620 Mpa (ZrO ball of 3.9 mm on cotanct with TiN surfaces) [41-43]. All the tests were carried out at room temperature. The wear tracks were analyzed using optical microscopy and an optical profilometer. The wear rate was calculated using:
Where V is the wear volume (ASTM 133), L is the sliding distance and F is the applied force [44].
- The figure 2a includes many information, it is recommended to improve the XRD plots for better readability.
The Figure 2a was modified to:
- While TiCrN’s higher hardness but lower corrosion resistance. However, the authors do not address why TiCrN underperforms electrochemically despite Cr’s known corrosion resistance!
This text was added to the work:
The addition of carbon (C) and chromium (Cr) has been utilized to enhance the mechani-cal and electrochemical properties of materials, particularly due to Cr. The inclusion of chromium can lead to the formation of a Cr2O3 film, which exhibits high chemical and mechanical stability [45, 77-79]. However, these findings indicate a decrease in the corro-sion resistance of the TiN film when C and Cr are added. This reduction can be attributed to the increased filtration of corrosive solutions through the porosity and defects present in the TiN matrix, which are caused by macroparticles and variations in the composition generated during the deposition process [73, 78-80].
- A critical question is related to the obtained Hardness of the TiN, TiCN and TiCrN, since the thicknesses are (TiN coating: 1.9 μm TiCN coating: 2.5 μm and TiCrN coating: 2.15 μm). First, the authors have used several loads (i.e., 300g, 100g, 50g, and 25g). How can the authors be sure about the effect of the substrate on the resulting measurements of the hardness, as the coatings thickness is round ~ 2 μm? Second, there is hardness anomaly, could the authors explain why do TiN and TiCN hardness values (12.2 GPa and 12.1 GPa) deviate from literature norms (~20-25 GPa)?
This text was added to the work:
Although, the reported value of the harness of the TiN films are from 13.3 GPa [57], and variety with the film thickness [58], the lower hardness value of the TiN and TiCN films than the reported could be produced by the substrate effect due to effects of the plastic de-formation of the substrate, causing that the hardness measurement were a composite hardness [59].
- Can the authors explain why does TiCrN exhibit a negative protection percentage (-41.23%, Table 1)? Is this due to CrN phase instability or coating defects, and how was this verified?
This text was added to the work:
This effect can be observed in the variation of the Pi, where the TiCN and TiCrN films pre-sented a lower Pi value than the TiN film, especially the Pi value of the TiCrN film, where the increment of the area improves the Redox process, causing a higher ion liberation that produced a higher icorr on the coated D2 steel surfaces with TiCrN film than the Uncoated D2 surfaces

Round 2
Reviewer 1 Report
Comments and Suggestions for Authors
It is suggested to establish a quantitative relationship between preparation technology and design development. Compared with other works, such as Journal of Materials Research and Technology 2024 (https://doi.org/10.1016/j.jmrt.2024.11.024).
Author Response
Report to reviewer 1
It is suggested to establish a quantitative relationship between preparation technology and design development. Compared with other works, such as Journal of Materials Research and Technology 2024 (https://doi.org/10.1016/j.jmrt.2024.11.024).
This work shows the quantitative relationship between deposition process parame-ters and coating performance. The development of the coating by PVD-Cathodic Arc tech-nology influences the mechanical, electrochemical and tribological properties. The quan-titative results between the deposition parameters and the resulting properties of TiN, TiCN and TiCrN coatings on D2 steel show the correlations in the manuscript data. This will allow to design coatings suitable for particular industrial environments as described by Xian et al. [42] and Du et al. [43].

Reviewer 2 Report
Comments and Suggestions for Authors
The authors have partially addressed the reviewer's comments. However, the response to this comment is unsatisfactory:
- A critical question is related to the obtained Hardness of the TiN, TiCN and TiCrN, since the thicknesses are (TiN coating: 1.9 μm TiCN coating: 2.5 μm and TiCrN coating: 2.15 μm). First, the authors have used several loads (i.e., 300g, 100g, 50g, and 25g). How can the authors be sure about the effect of the substrate on the resulting measurements of the hardness, as the coatings thickness is round ~ 2 μm? Second, there is hardness anomaly, could the authors explain why do TiN and TiCN hardness values (12.2 GPa and 12.1 GPa) deviate from literature norms (~20-25 GPa)?
This is your response.
"This text was added to the work:
Although, the reported value of the harness of the TiN films are from 13.3 GPa [57], and variety with the film thickness [58], the lower hardness value of the TiN and TiCN films than the reported could be produced by the substrate effect due to effects of the plastic de-formation of the substrate, causing that the hardness measurement were a composite hardness [59]."
However, If the effect of the substrate is acounted in your measurement of the hardness, then the reported values are questionable!! That means the authors did not adequately take the necessary cautions to measure the coatings hardness!!!
Can the authors measured again the harness of their samples using instrumented nanoindentation, a means that can reduce the substrate effects on the hardness of coatings by monitiring the used loads and and the indenter depth?
Author Response
Report to reviewer 2
The authors have partially addressed the reviewer's comments. However, the response to this comment is unsatisfactory:
A critical question is related to the obtained Hardness of the TiN, TiCN and TiCrN, since the thicknesses are (TiN coating: 1.9 μm TiCN coating: 2.5 μm and TiCrN coating: 2.15 μm). First, the authors have used several loads (i.e., 300g, 100g, 50g, and 25g). How can the authors be sure about the effect of the substrate on the resulting measurements of the hardness, as the coatings thickness is round ~ 2 μm? Second, there is hardness anomaly, could the authors explain why do TiN and TiCN hardness values (12.2 GPa and 12.1 GPa) deviate from literature norms (~20-25 GPa)?
This is your response.
"This text was added to the work:
Although, the reported value of the harness of the TiN films are from 13.3 GPa [57], and variety with the film thickness [58], the lower hardness value of the TiN and TiCN films than the reported could be produced by the substrate effect due to effects of the plastic de-formation of the substrate, causing that the hardness measurement were a composite hardness [59]."
However, If the effect of the substrate is acounted in your measurement of the hardness, then the reported values are questionable!! That means the authors did not adequately take the necessary cautions to measure the coatings hardness!!!
Can the authors measured again the harness of their samples using instrumented nanoindentation, a means that can reduce the substrate effects on the hardness of coatings by monitiring the used loads and and the indenter depth?
Unfortunately, we do not have access to a nanoindenter system to measure mechanical properties of the films in just 5 days. Nevertheless, the hardness measured with microindentation Vickers tests to update the hardness value of the film and the hardness graphic. In the same way, theoretical hardness of the films was determinate using Korsunsky mathematical model. Whit this, the next text and graphic were included.
The hardness values measured for the coatings TiN, TiCN, and TiCrN were 16.1 ± 2.7 GPa, 14.2 ± 1.2 GPa, and 18.6 ± 3.7 GPa, respectively. While these values are comparable to some previously reported hardness values for TiN films, they fall within the lower range of those reported for TiN [63-67]. This variation can be attributed to the fact that during the application of load, the contact stresses exceed the thickness of the films. This leads to a combination of plastic deformation of the coating and material removal. Evidence of this plastic deformation was observed through the contact depth values obtained during the microhardness Vickers indentation test at a load of 25 g, where the contact depths were found to be 0.76 ± 0.06 μm, 0.8 ± 0.02 μm, and 0.71 ± 0.08 μm for the TiN, TiCN, and TiCrN films, respectively, exceeding more than 10% of the films’ thickness [68-70]. In order to obtain a theoretical approximation value of the films was used the Korsunsky mathematical model to determinate the hardness of the films [71-73]:
H_c=H_s+(H_f-H_s)/(1+kβ^2 )
Where: Hc is the composite hardness, Hf is film hardness, Hs is the substrate hardness, k is afitting parameter (in this case was used the k value reported by Tuck et al. [71] for the TiN film deposited by catholic arc technique) and β is defined by Korsunsky et al. [72] as the indentation depth normalized with respect to the coating thickness.
The hardness value obtained with the model were 31.9±5.2 GPa, 23.2± 2.7 and 32.2±5.7 GPa for TiN, TiCN and TiCrN films, respectively. These hardness value are more similar to the reported hardness value for TiN film [65, 71, 74, 75].
